# Pharmacists and Contraception in the Inpatient Setting

**DOI:** 10.3390/pharmacy8020082

**Published:** 2020-05-09

**Authors:** Domenique Ciriello, Nicole Cieri-Hutcherson

**Affiliations:** 1Kaleida Health/Buffalo General Medical Center, Buffalo, NY 14203, USA; dciriello@kaleidahealth.org; 2School of Pharmacy and Pharmaceutical Sciences, University at Buffalo, Buffalo, NY 14260, USA

**Keywords:** inpatient, hospital, contraception, pharmacist, counseling, REMS, contraindication, teratogenic

## Abstract

The choice of contraceptive method should be based on patient specific factors, patient preference, and method-specific properties. In this article, we review opportunities for an inpatient clinical pharmacist to assist in the selection and counseling of contraceptives in hospitalized patients. An inpatient pharmacist has the opportunity to discuss various contraceptive methods with the patient, ensuring an appropriate method is used after discharge, which is especially important after the occurrence of a contraception-related adverse effect or contraindication to certain contraceptive methods. Barriers, such as formulary restrictions, can limit inpatient initiation of contraceptive therapy while hospitalized, but pharmacists can provide education on appropriate alternatives. Inpatient clinical pharmacists can also make recommendations for contraceptive methods in special populations. It is crucial to select an appropriate therapy in patients with an underlying medical condition, such as those with active or history of breast cancer, psychiatric disorder, or thrombophilia, as inappropriate therapy can cause an increased risk of harm. Pharmacists can assist in contraceptive counseling, evaluating for drug-drug and drug-disease interactions, and recommending the most appropriate therapy in special populations. An inpatient pharmacist has the opportunity to interact with the medical team and assist in navigation of teratogenic medication use and Risk Evaluation and Mitigation Strategies.

## 1. Introduction

Unintended pregnancy rates in the United States (US) have been reported to be as high as 45%, and can place a burden on parents, children, caregivers, and the healthcare system [1,2]. Inappropriate use of contraception may contribute to the high levels of unintended pregnancies in the US [2]. It is estimated that 50% of women who experience an unintended pregnancy were not on contraceptive therapy at the time of conception, while the other half become pregnant due to inconsistent or incorrect use of contraceptives [3]. Up to 120,000 infants are born with some form of physical or mental birth defect per year [4]. For most birth defects, the cause is unknown; however, birth defects may be caused by smoking, alcohol use, chronic and acute medical conditions, and certain medications [4]. Approximately 6% of pregnancies in the US are exposed to a potentially teratogenic medication, and 3% of pregnancies in the US result in a physical or mental birth defect [5]. Patients of reproductive age maintained on a teratogenic medication should be advised of the risk of the teratogenic medication and offered an appropriate contraceptive method. A Risk Evaluation and Mitigation Strategy (REMS) is a drug monitoring program that the US Food and Drug Administration (FDA) mandates for providers, pharmacies, patients and/or manufacturers to ensure the benefits of a medication outweigh the risks. Not all teratogenic medications require a REMS, and not all REMS are associated with teratogenic medications [6]. REMS may include a risk mitigation goal and additional medication-specific requirements, such as enrolling in a specific monitoring program, undergoing laboratory testing, and reporting medication use and errors to MedWatch [6].

Hospitalized patients of childbearing potential may be initiated on a teratogenic medication, presenting an opportunity for pharmacist intervention. Examples of teratogenic medications that could be initiated in hospitalized patients are listed in Section 5. Recommendations can be made for appropriate contraceptive therapy to prevent an unintended pregnancy and to mitigate the risks of the teratogenic medication. Patients may also present to the hospital with adverse effects associated with a contraceptive or a new disease state, which may warrant further counseling on contraceptive methods. While hospitalized, a clinical pharmacist can assist in making recommendations for appropriate contraceptive therapy, counseling the patient on the contraceptive therapy and/or the teratogenic medication, and assisting providers and patients in navigating REMS.

## 2. Contraindications and Adverse Effects Associated with Hormonal Contraceptives

The most commonly prescribed form of contraception in the US is oral contraception, and the most commonly prescribed oral method is the combined hormonal contraceptive tablet [7]. Combined hormonal contraceptives (CHC), which contain both an estrogen and progestin, may be associated with serious adverse reactions related to one or both components. An overview of various contraceptive methods and hormone component(s) is available in Table 1. Progestins are associated with increased risks of acne, hirsutism, and breakthrough bleeding. These adverse effects are usually considered minor and may be mitigated by increasing or decreasing the progestin component in the CHC, switching to a CHC with different androgenicity, or switching to a non-hormonal contraceptive [8]. Estrogens are associated with increased risks of nausea, headache, breakthrough bleeding, and breast tenderness [9]. Patients experiencing a minor adverse effect related to the estrogen may benefit from alterations (increasing or decreasing) in the estrogen component of the CHC, switching to a progestin-only contraceptive, or switching to a non-hormonal contraceptive [8,9].

The Center for Disease Control and Prevention’s (CDC) Summary of Medical Eligibility Criteria for Contraceptive (US-MEC) discusses in detail the appropriateness of contraceptive therapies based on various medical conditions [10]. Table 2 is a list of absolute or relative contraindications to CHC therapy based on the hormone component as outlined in the CDC-MEC [9,10]. Some adverse reactions attributed to CHC may be severe, causing patients to discontinue them permanently; additionally, most contraindications to CHC can be attributed to the estrogen component [9,10]. Patients are at risk of breast cancer and liver cancer while on estrogen-containing contraception; however, they are at a reduced risk of ovarian, endometrial, and colorectal cancer [10]. Postmenopausal women treated with oral estrogen therapy combined with medroxyprogesterone are at an increased risk of invasive breast cancer, compared to premenopausal woman [10]. Postmenopausal women treated with estrogen (with or without progestin) are at an increased risk of adverse effects, specifically myocardial infarction, stroke, pulmonary emboli, and deep vein thrombosis, compared to premenopausal women [10]. While CHC have adverse effects associated with use, they can be advantageous when used in the appropriate patient population. Therefore, recommending an appropriate contraceptive method to mitigate adverse reactions is imperative. Appropriate patient education and prioritization of patient input allows women to be knowledgeable in their contraceptive choices, leading to appropriate use and compliance over time [11].

If a patient is hospitalized with an adverse reaction or contraindication to CHC, emergent discontinuation of the current contraceptive is usually warranted. Patients maintained on CHC could be hospitalized for development of a venous thromboembolism (VTE), which would warrant immediate discontinuation of the CHC. While the overall incidence of CHC associated VTE is low (1-5/10,000 women years), incidence varies by hormone component [12]. Patients initiated on a drospirenone-containing CHC had a 77% increased risk of hospitalization related to VTE compared to low-dose estrogen CHC comparators [13]. Although patients admitted with an adverse reaction to CHC often warrants discontinuation of the offending agent, this event creates opportunities for pharmacists to discuss a medication therapy more tailored to the patient and their new disease state. A clinical pharmacist may play a significant role in recommending an appropriate alternative and providing patient education to improve contraceptive adherence. When switching between contraceptive methods, clinical pharmacists can educate patients on appropriate use of back-up methods, management of missed doses, mitigation of adverse effects, formulation-specific administration, and clinical pearls [14].

## 3. Barriers and Facilitators to Initiating Contraceptive Therapy in an Inpatient Setting

Contraceptive therapy is often discontinued in hospitalized patients, including those with an adverse reaction or contraindication to contraceptive therapy; however, patients are rarely initiated on a more appropriate contraceptive therapy despite potential benefits. Inpatient prescribers may feel uncomfortable initiating contraceptive therapy or deem it non-essential, deferring the decision until outpatient follow-up. Hormonal contraceptive therapy may also be held due to the increased risk of VTE, which can be a concern in hospitalized patients [15]. Hospital formulary limitations may also be a substantial barrier to initiating contraceptive therapy. In the instance that contraceptive therapies are on a hospital formulary, often only one type of hormonal contraceptive, such as a CHC tablet, is available [16].

If contraceptives were to be available to hospitalized patients, select patients may benefit from initiating therapy. Non-contraceptive benefits for patients may include improvement of symptoms of menstrual cycle disorders such as dysmenorrhea, chronic pelvic pain, menorrhagia, and endometriosis. [17]. Inpatients presenting with these symptoms may be initiated on appropriate contraceptives. Hospitalized women who wish to prevent unintended pregnancies may benefit from initiation of contraceptive therapy while inpatient. Pharmacists can advocate for addition of select contraceptives on formularies for these indications.

Inpatient initiation of contraceptives are often limited to insertion of a long-acting reversible contraceptive (LARC) in the immediate postpartum period. Intrauterine devices (IUD) and the progestin-only implant are the most commonly used methods of postpartum LARC [18]. Postpartum LARC therapy has become increasingly used after childbirth and has been found to reduce barriers to contraceptive access encountered after childbirth, including loss to follow-up or potentially the loss of insurance coverage [18,19]. Up to 40% of postpartum patients do not attend follow-up appointments, and 75% of women who plan to get a postpartum IUD do not obtain it [18]. Inpatient insertion could mitigate these issues. Postpartum LARC therapy is efficacious and cost-effective [18,19]. An increased risk of device expulsion and additional contraindications to therapy (intrauterine infection at time of delivery, postpartum hemorrhage, and puerperal sepsis) are limitations to IUD therapy [18]. No additional contraindications or risks are associated with the contraceptive implant other than those listed in Table 2 [18].

While initiation of contraceptive therapies may be limited to certain hospitalized patients, inpatient pharmacists can still assist in patient and provider education for contraception initiation after discharge. Education provided to the patient regarding a safer, more appropriate, contraceptive therapy could motivate the patient to ensure adequate follow-up upon discharge. Pharmacists can assist in patient education and counseling for patients considering or receiving postpartum LARC. Patient education can include the advantages therapy, including the high efficacy rate, and the convenience of initiating therapy while hospitalized [18,19]. Pharmacists can educate the patient on adverse reactions of therapy and provide information regarding alternative therapies.

## 4. Special Populations

Selection of an appropriate contraceptive for all patients is important but especially crucial in patients who have an underlying medical condition where inappropriate use could put the patient at increased risk of harm. Appropriate contraceptive counseling and selection of an optimal contraceptive can increase medication compliance, reduce risks associated with the medications, and improve patient outcomes [20]. Contraception use in select medical conditions may include, but are not limited to, active or history of breast cancer, psychiatric disorders, and thrombophilia. Understanding the potential risks associated with a particular disease state and the contraceptives available are crucial.

Breast cancer is more commonly diagnosed in women over 50 years old, but is still prevalent in younger women. In the US, approximately 11% of new cases of breast cancer are diagnosed in women less than 45 years old, indicating contraceptive therapy may be warranted in these patients due to reproductive potential [21]. Estrogen may stimulate growth of certain breast cancers and the role of progestin remains controversial [22]. A large cohort of women who recently used or were currently on any hormonal contraceptives were found to have an increased risk of breast cancer compared to women who had never used hormonal contraceptives [23]. An increased risk of breast cancer among patients using progestin-only contraceptives was observed, but the risks were inconsistent between various formulations [23]. Dose-response and duration-response relationships were absent, making the association difficult to interpret [23]. The US-MEC recommends against hormonal contraceptives in patients with active breast cancer [10]. The only contraceptive therapy recommended without restriction in patients with active breast cancer is the copper-containing IUD [11]. Use of hormonal contraceptives in patients without active disease but a history of breast cancer is also not recommended by the US-MEC as the theoretical risk of breast cancer returning outweighs the benefits [11]. The copper-containing IUD is the only option without restriction [11]. Although risks with estrogen therapy and breast cancer are well known, patients with a familial history or with breast cancer susceptibility genes (BRCA1 and BRCA2) do not have an additional risk with CHC [10,24,25,26,27]. Therefore, all contraceptive therapies available are considered safe, potential options [10]. If a patient with a family or personal history of breast cancer is encountered while hospitalized, a pharmacist can serve as a resource to determine the risks associated with each contraceptive therapy in order to recommend an appropriate therapy upon discharge.

There are also many considerations when recommending contraceptive therapy in women with psychiatric disorders. Women of reproductive age are at the highest risk of developing a major depressive disorder [20]. Contraceptive choices for these patients can be challenging for health care providers due to factors such as drug-drug interactions, the possibility of mood changes with hormonal contraceptives, and compliance [20]. These patients may also be at risk of teratogenic effects to a fetus from the medications initiated for management of psychiatric illnesses without reliable contraception [28]. Carbamazepine may be used in women with a psychiatric illness and may diminish the effects of CHC due to enzyme induction [20]. Lithium does not pose a drug interaction with contraceptive therapy, but ideally should be avoided in pregnancy, especially in the first trimester due to teratogenic potential [20]. Valproic acid, which is often used as a mood stabilizer, should be avoided if possible in pregnancy due to concerns of major congenital malformations, particularly neural tube defects [29]. Serotonin reuptake inhibitors (SSRIs) and serotonin and noradrenaline reuptake inhibitors (SNRIs) are commonly used for psychiatric disorders such as depressive disorder and anxiety disorders [30]. Risks in pregnancy for SSRIs and SNRIs are not class-wide, therefore each medication should be evaluated for teratogenic potential [30]. For example, paroxetine, an SSRI, is not usually recommended due to risks of congenital heart disorders [30]. Unintended pregnancies may pose added challenges for women with a psychiatric disorder and could cause a relapse in depression and other psychiatric illnesses [20]. Increasing data suggests that mental illness and substance abuse disorders are risk factors for nonuse and noncompliance with contraceptives [31,32,33,34,35]. The US-MEC classifies depression as a medical condition without restrictions to the use of any contraceptive options and does not make any specific recommendations for other psychiatric disorders such as schizophrenia, bipolar disorder, or postpartum depression [10]. Alterations in mood from hormonal therapy are a concern in these patients, as it can worsen the disease. In a systematic review including patients with depression and bipolar disorder, patients were not at an increased risk of worse clinical course of illness when maintained on CHC tablets, progestin-only IUDs, or progestin-only injectables compared to no hormonal therapy [28]. Additionally, compliance may be difficult in patients experiencing psychiatric disorders. A retrospective review with over 9000 patients investigated contraceptive adherence in patients with depression, posttraumatic stress disorder, anxiety, bipolar disorder, schizophrenia, or adjustment disorder, with or without substance abuse disorder [35]. When compared to women without psychiatric disorders, women with psychiatric disorders on any hormonal contraceptives were more likely to have reduced contraception adherence and continuation [35]. Reduced adherence rates and discontinuation rates were even higher among women with a psychiatric and substance abuse disorder [35]. An IUD may be a potential option for these patients if adherence is an issue and preventing pregnancy is the major concern. In a cohort including women with bipolar disorder, increased compliance rates at one year were higher among women using either a copper-containing or progestin-only IUD as contraceptives compared to women using a progestin-only injection [36]. If a patient with a psychiatric disorder is encountered while hospitalized, a clinical pharmacist can assist in several ways to select an appropriate therapy. A pharmacist can be used to research the literature available for contraceptive management with the specific psychiatric disorder, evaluate for drug-drug interactions with medications for the psychiatric illness, and recommend a regimen to assure compliance upon discharge.

Women with thrombophilia such as Factor V Leiden, prothrombin mutation, and protein S, protein C, or antithrombin deficiencies, could have a two- to twenty-fold higher risk of thrombosis if maintained on CHC compared to women not on CHC [10]. The US-MEC classifies CHC as unacceptable methods for patients with a known thromboembolic disorder [10]. Progestin-only contraceptives are recommended by the US-MEC as advantages generally outweigh the theoretical risks [10]. A copper-containing IUD contributes no additional VTE risk [10]. Additionally, thromboembolic risk increases with increasing estrogen dose; hence, newer oral CHC formulations often contain a reduced dose (between 15 and 35 μg of ethinyl estradiol) [37]. Thromboembolic risk is also dependent upon progestin type [37,38]. A CHC containing a third or fourth generation progestin, such as drospirenone, has a higher thromboembolic risk compared to first or second generation progestins, such as norethynodrel [37,38]. Non-oral administration of contraceptives may result in increased estrogen exposure due to avoidance of first-pass effect on metabolism [39]. Studies suggested that the CHC patch has an increased risk of VTE given the increased exposure to estrogen [39,40,41,42]. Studies investigating the prothrombic effect of the CHC vaginal ring showed conflicting results [39,40,41]. The absolute risk of VTE in patients on CHC with Factor V Leiden was found to be lower than the risk associated with pregnancy and the post-partum period [43]. Universal screening for thromboembolic conditions is not recommended prior to prescribing CHC unless patients have a family history of VTE in a first-degree relative in a woman of a reproductive age [37,43,44]. Screening based on VTE history was also shown to be more cost-effective than universal screening [45]. If patients cannot tolerate a CHC containing a first or second generation progestin, it is unclear whether screening for thromboembolism disorders should be performed prior to initiating a CHC with a third or fourth generation progestin [37]. An inpatient pharmacist can assist in recommending an appropriate therapy based on route of administration, estrogen dose and generation of progestin for initiation after discharge. Pharmacists can also evaluate patients for screening for thromboembolic conditions is appropriate prior to hormonal contraception initiation.

## 5. Counseling on Contraception when Initiating Teratogenic Medications and Navigating REMS

Patients of reproductive age may be initiated on a teratogenic medication while hospitalized or prior to admission. Teratogenic medications should ideally be avoided in patients of reproductive age; however, the benefits of the teratogenic agent may outweigh the risks. Teratogenic medications more commonly used include: antiepileptic medications, renin-angiotensin-aldosterone system inhibitors, 3-hydroxy-3-methyl-glutaryl-CoA reductase inhibitors, tetracyclines, warfarin, methimazole, lithium, non-steroidal anti-inflammatory drugs, ciprofloxacin, sulfonamides, and methotrexate [46].

Inadequate outpatient counseling on teratogenic medications may lead to medication misuse [47]. Challenges that primary care physicians may encounter when appropriately counseling on teratogenic medications include lack of time, lack of reimbursement, difficulty finding relevant information, and difficulty identifying patients’ pregnancy plans [47]. Some of these barriers to counseling may also exist for inpatient pharmacists. In the face of these hurdles, it is important to note that counseling patients may be effective in preventing associated risk. A survey conducted with 800 patients in primary care clinics concluded that women who received adequate counseling about contraception and teratogenic risks were more likely to use contraceptives compared to women who received no counseling [47]. This reinforces the necessity for patient counseling, regardless of whether patients are hospitalized or seen in clinic. Counseling by a health care professional regarding the teratogenic medications and contraception is important because studies showed that patients commonly sought information from friends or relatives, which may not relay reliable information [5,48]. In a qualitative study, women felt that detailed information regarding pregnancy-related risks were necessary to prevent unfavorable outcomes [5]. As the experts in medication therapy, pharmacists can play an essential role in medication counseling to ensure patients receive appropriate and accurate information. This can prevent medication misuse and prevent patients from potentially seeking information from unreliable sources. If a patient is initiated on a teratogenic medication while hospitalized, an inpatient clinical pharmacist can assist in counseling the patient regarding the risks and benefits of the medication as well as appropriate contraceptive use to prevent pregnancy [49]. Advanced drug-pregnancy alerts can be implemented by organizations to alert inpatient pharmacists when hospitalized patients may be or are pregnant, and these may be particularly helpful when deciding whether to initiate teratogenic medications. Automated alerts are not without errors. A pregnancy test may not be performed on admission, and additionally, if performed, a negative pregnancy test may not guarantee the absence of pregnancy (i.e., if performed too early to detect or as part of a previous recent admission). Systems may not be updated when a patient is no longer pregnant, thus providing inaccurate information about patient status. Some institutions implemented an automated trigger for specific teratogenic medications deemed catastrophic if administered to pregnant women [50]. Drug compendia may provide guidance on counseling and important precautions for females of reproductive potential initiated on potentially teratogenic medications [51].

Contraceptive use is influenced by a variety of factors including, but not limited to: ethnic differences, religious beliefs, racial differences, and access to medical care [11]. Initiation of contraceptive counseling during an inpatient stay may reduce the risk of an unintended pregnancy [11]. Optimized contraceptive counseling has been associated with positive outcomes, including increased adherence, reduction in adverse events, and positive relationships between patients and their providers [11]. Relational communication and task-oriented communication are supported in the literature as two methods to successfully counsel patients on contraceptives [12]. Relational communication focuses on a positive relationship between health care professionals and patients [11]. Components of positive relational communication include building trust with patients, addressing patients concerns regarding contraceptive therapy, and using a shared decision making by centering patient preferences for contraceptive methods [11]. Task-oriented communication includes delivering necessary information regarding diagnoses and treatment options [11]. Effective task-oriented communication includes offering sufficient counseling on adverse reactions and the necessity for dual protection for women at risk of sexually transmitted infections [11]. Barriers and misperceptions regarding contraceptive use should be addressed to ensure consistent and correct contraceptive use [11]. Limited time in the inpatient setting may make these communication strategies more challenging; nonetheless, counseling for optimal contraceptive use remains the desired outcome.

Pharmacists may be used as a resource by health care professionals and patients regarding the navigation of REMS. Patients initiated on teratogenic medications may involve requirements of a REMS. Most REMS have a communication component regarding specific safety concerns [52]. Other medications may have additional requirements for patients or health care providers that can involve enrolling in a specific monitoring program, undergoing laboratory testing, and reporting medication use and errors to MedWatch [52]. A list of various requirements for patients, health care providers, pharmacists/pharmacies, and manufacturers are available in Table 3 [53]. The FDA has over 50 REMS for various medications, which can be overwhelming to navigate if unfamiliar with a specific medication [54]. Each REMS has different goals that are outlined with each medication. Not all REMS are related to teratogenic potential; some were implemented to monitor for other serious adverse effects (i.e., clozapine for the risk of severe neutropenia or vandetanib for the serious risks of QT prolongation, Torsade de pointes, and sudden death). Examples of REMS instituted to mitigate teratogenic risks of teratogenic potential include mycophenolate, isotretinoin, and thalidomide. The FDA provides various materials such as DVDs, video and paper instructions, patient comprehension questions, pharmacist’s guides, and REMS documents to provide education to all participants involved.

## 6. Conclusions

A clinical pharmacist can play an essential role in counseling on contraceptive options following the occurrence of adverse effects and contraindications, counseling on teratogenic medications and contraceptive options to mitigate risk, navigation of REMS programs specifically relating to females of reproductive potential, and contraceptive choices for special populations encountered by the hospital. The US-MEC offers recommendations and resources for patients and health care providers for selecting an appropriate contraceptive therapy based on various medical conditions.

## Figures and Tables

**Table 1 pharmacy-08-00082-t001:** Contraceptive Formulation by Hormone Component [10].

Hormone Component	Type of Contraceptive Available
Combined Hormonal Contraceptive (Estrogen + Progestin)	Tablet Patch Vaginal Ring
Progestin-only	Tablet Implant Intrauterine Device Injection

**Table 2 pharmacy-08-00082-t002:** Absolute or relative contraindications to CHC based on hormone component [9,10].

Hormone Component	Risks
Combined Hormonal Contraceptive (Estrogen + Progestin)	Current breast cancer
	Breastfeeding less than 21 days postpartum
	Severe decompensated cirrhosis
	Acute venous thromboembolism
	High risk of recurrent venous thromboembolism
	Major surgery with prolonged immobilization
	Complicated diabetes
	Migraine headache with aura
	Uncontrolled hypertension
	Current of history of ischemic heart disease
	Known thrombogenic mutations (i.e., Factor V Leiden; prothrombin mutation; and protein S, protein C, and antithrombin deficiencies)
	Multiple risk factors for atherosclerotic cardiovascular disease
	Peripartum cardiomyopathy
	Smokers greater than 35 years old
	History of cerebrovascular accident
	Systemic lupus erythematosus with positive antiphospholipid antibodies
	Complicated valvular heart disease
Progestin-only	Current breast cancer
	Severe decompensated cirrhosis
	Systemic lupus erythematosus with positive antiphospholipid antibodies

**Table 3 pharmacy-08-00082-t003:** Potential REMS Requirements and Resources [53].

Patients and Caregivers	Requirements could include: Sign a form acknowledging understanding of risks before initiating the medication Undergo laboratory testing Enroll in a registry to monitor or document adverse events Resources: Talk to HCP Contact the FDA Contact REMS Form—https://www.accessdata.fda.gov/scripts/email/cder/comment-rems.cfm Contact the REMS program directly
HCP who prescribe medications	Requirements could include: Complete training Enroll in the REMS Document counseling of patients Enroll patients Monitor and/document compliance with certain safe use conditions Report drug errors, adverse events, or other safety aspects related to REMS medications to MedWatch Resource for HCP: FDA: https://www.accessdata.fda.gov/scripts/cder/rems/index.cfm
Pharmacies or Health Care Settings that dispense Medications with REMS	Pharmacy requirements could include: Certification to dispense the medication Pharmacist requirements may include: Completion of required training Verify safe use conditions Provide patient education Provide patients with educations materials or medication guides Report drug errors, adverse events, or other safety aspects related to REMS medications to MedWatch Resources for pharmacists: FDA: https://www.accessdata.fda.gov/scripts/cder/rems/index.cfm Product labeling REMS specific websites
Drug Manufacturers	Plan, develop, submit for approval, implement, assess, and/or modify REMS Guidelines for manufacturers to follow and review: Revised Draft Guidance for Industry on Format and Content of a Risk Evaluation and Mitigation Strategy Document Guidance for Industry: Use of a Drug Master File (DMF) for Shared System REMS Submissions Risk Evaluation and Mitigation Strategies: Modifications and Revisions—Guidance for Industry FDA’s Application of Statutory Factors in Determining When a REMS is Necessary

Key: DMF: Drug Master File; FDA: Food and Drug Administration; HCP: Health Care Professional; REMS: Risk Evaluation and Mitigation Strategy.

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
