# Peer review of "Pharmacists and Contraception in the Inpatient Setting"

_pharmacy, 2020, doi:10.3390/pharmacy8020082_

Round 1

Reviewer 1 Report

Introduction:

Line 30-31: The wording of this sentence should be revised as the current phrasing makes it seem that contraceptives are blocking the teratogenic effect of the medication, not preventing the pregnancy (therefore no teratogenic effect can occur).

Line 40-41: Please add specific examples of medication-specific requirements included in REMS.

Line 42: Please add specific examples of "common" teratogenic medications that are initiated in hospitalized patients. Additional, some information regarding the frequency or extent of initiation of these types of medications would help the reader better judge the impact of a pharmacist in the inpatient setting. 

Line 44: Please add "teratogenic" between "... risks of the" & "medication".

Line 46: Remove "...and the availability of future contraceptive methods"

Overall: Consider discussing the impact of hospital formularies on medication choice or the ability to have a LARC inserted during a hospital stay unrelated to contraception.

Section 2:

Overall: This section heading makes the reader feel that they will be receiving a "guide" for counseling tips. Consider revising the section heading to more accurately reflect the included content. "Progestin" and "progesterone" are used interchangeably throughout this section in particular. Also switch between CHC and CHCs. Please be consistent.

Line 59: Remove "... depending on the adverse effect" 

Line 65-73: This list is exhaustive and difficult to read. Suggest limiting this list or creating a table with this information instead. 

Line 75: Remove "... in combined hormonal contraceptives".

Line 77-79: Please add a description of who is at risk for adverse effects to make this section more applicable to readers. Additionally, this statement is repetitive with the statement in lines 61-64. 

Line 89: Please add "providing" before "...patient education to improve..."

Section 3:

Line 94: Remove "a" from "Patients of a reproductive age"

Line 103-108: This section related to primary care information is irrelevant to inpatient pharmacists (and is the only data presented like this). Consider removing or more directly linking to inpatient pharmacists.

Line 108-110: Consider rewording this sentence as it is awkward to read through.

Line 115-117: Can you provide any recommendations or evidence about treatment algorithms or use of "automatic triggers" to alert pharmacists when certain medications are initiated on an admitted patient?

Line 119: This is the first mention of "males" in the manuscript. Would suggest either more fully developing the concept of male contraception throughout the manuscript, or eliminating it.

Line 120: Also consider religious views

Line 121-122: Will all patients feel it is a "positive influence" to have a pharmacist talking to them about contraception when they are admitted to the hospital and started on a medication for an acute problem?

Line 122: Change contraceptive to "contraception" or "contraceptives"

Line 127-130: Are inpatient pharmacists truly able to create these relationships for communication when there is such a focus on discharging as soon as medically safe?

Table 2:

  • Patients and caregivers: Should there be a ) at the end of the link?
  • HCP who prescribe medications: Change to "Enroll" and "Monitor" to maintain consistency with other words in table
  • Pharmacies or Health Care Settings: In Column 2 --> Change "Pharmacies" to "Pharmacy". Remove "Individual"

Section 4:

Line 163: Reword following comma. "...,where contraceptive therapy may be warranted due to reproductive potential."

Line 165: Change included to "found that"

Line 167-169: What forms are associated with higher risks?

Line 170-172: How does a risk of breast cancer associated with product use matter for someone that already has active breast cancer?

Line 173-175: The theoretical risks of what outweigh the benefits?

Line 197: Delete "concluded"

Line 198: Change "in patients" to "when"

Line 199: Delete "when"

Line 203-204: Please add what methods were included

Line 205-206: Revise this sentence. It does not make sense as written currently... "reduced rates were higher?"

Line 206-207: Concern related to the assumption that pregnancy prevention is the number one priority for people with adherence issues. 

Line 211-214: While I agree that this is an important aspect, the common drug-drug interactions were not discussed. Please add examples of specific common interactions.

Line 221-222: Are the estrogen levels included here higher or lower than the older forms that are referred to?

Line 224: Change "have" to "has"

Line 229-230: Please clarify if this refers to when using CHCs

Conclusion:

Overall, I feel that this manuscript needs to be strengthened to include how often these types of inpatient admissions are occurring. Please include more data about the potential impact and new opportunities that would be created with a shift for inpatient pharmacists in this way.

Reviewer 2 Report

Overall comment: Thank you for the review of the pharmacist's role in contraceptive services related to contraceptive selection, navigating teratogenic medications and REMS, and special populations. However, I feel that there was little information that related this to what an inpatient pharmacist's role may be and that the commentary provided could be applicable in both the inpatient and outpatient setting. Are there barriers to contraceptive access in the inpatient setting, be it due to hospital policy (no contraception on formulary but patients can bring in their own) or due to hospital policies (in my catholic institution, IUDs cannot be prescribed/inserted). What counseling should be done if patients do not immeditaely resume birth control upon hospitalization? Are there any procedures for which birth control should be held? During my inpatient rotation experiences, initiating contraception was never discussed as patients were usually referred to their primary care providers. Are there certain settings in which the authors feel that inpatient intervention would be beneficial (ex - internal medicine team, OB-GYN floor, other subspecialties, etc).

23. Consider removing 'the' before parents, children, etc.

24. Consider a statistics on the number of unintended pregnancies in the US vs intended pregnancies.

26-27. Sentence states that 50% of unintended pregnancies occur despite use of contraceptives. Can you elaborate? Is there literature discussing why this is (inappropriate use, ADR, etc)?  

37-41. Since teratogenic medications are mentioned just prior to this, I feel that this sentence reads as if all teratogenic medications are required to have a REMS. It is mentioned later, but consider a statement that REMS includes some teratogenic medications but this is not a requirement and that medications can have REMS for other reasons.

38. Food Drug administration should read 'Food and Drug administration', missing the 'and'

42. Consider adding 'of childbearing potential' to the phrase 'hospitalized patients'.

47. Consider removing 'an' before appropriate contraceptive

50-51, 152. I feel that information discussed in section 4 coincides with information in section 2. Section 3 discussing teratogens and REMS in the middle seems confusing. Consider switching section 3 and section 4.

52-53. Consider mentioning components of CHC (.. combined hormonal contraceptive, which contains estrogen and progesterone.)

53-55. Consider changing phrasing to 'CHC may be associated with one or both components' instead of not necessarily both components.

57-60. Consider mentioning an additional option when experiencing ADR to progestin as switching to different progestin with different androgenicity.

61-64. Is this sentence intended to discuss alternative options for both progestin/estrogen? Could consider altering phrasing to 'patients experiencing ADRs related to estrogen...'

65-84. Consider opening this paragraph discussing US-MEC. Moving sentence 79-82 'The CDC's MEC.... mitigate adverse reactions is imperative'. Could then follow with line 74-75 'Most contraindications to CHCs can be.. in CHC.'. Then listing situations where risk outweighs benefit.

65-73. This is a very long sentence. Consider listing in a table? Are these all related to both progestin and estrogen components or just one? The way it reads now appears it could be either, but as mentioned later in the paragraph its more likely to be related to estrogen. Could make a table discussing these risks and have two columns with estrogen/progestin.

89. Consider adding 'provide' before patient education

96-99. Consider RAAS agent instead of ACE since ARBs can be teratogenic. What about statins? doxycycline?

112. Consider changing 'expertise' to 'experts'.

112-115. Consider splitting into two sentences. 

136-149. I feel that this paragraph makes it appear that most REMS are for medications that are teratogenic. While some medications are (isotrentinoin, thalidomide), many have REMS for monitoring purposes (clozapine for agranulocytosis). I am not sure how much this contributes to literature regarding inpatient pharmacists contributing to the initiation of contraception. Consider the bigger focus being on the fact that REMS exists. Consider adding examples of teratogenic medications with REMS. 

Table 2: Tense varies throughout. 

Patients and caregivers: consider changing to 'sign a form acknowledging understanding of risks...'

HCP: Consider changing to 'enroll patients' and 'monitor/document...'.

Pharmacist/pharmacy: Consider changing to 'pharmacy requirements may include'; consider changing to 'completion of required training', consider 'provide patient education' instead of counsel patients

Drug manufacturers: consider switching verbs ending in -ing to present tense (plan, develop, submit, implement, etc).

152. As mentioned previously, this information appears to coincide with information in section 2. Could consider reformatting to include this information after section 2.

165-167. Consider rephrasing to 'a large cohort of women.... were found to have an increased...'

186. Drug-drug interactions with contraception and medications for mood are mentioned - can you list a few examples?

189-190. Teratogenicity with mood medications is mentioned, are there any medications that are of particular concern (ex - mood stabilizers)? What about our most commonly used agents (SSRI/SNRI/lamotrigine/atypical antipsychotics)

Round 2

Reviewer 1 Report

Thank you for the thoughtful revisions and effort put into revising your manuscript. 

Introduction:

  • Line 42-23: Please remove statement "Examples of teratogenic...". This would be more effective to list a couple of common specific agents, or simply remove this sentence altogether. If you choose to keep it, please update to reflect that Section 5 contains the referenced examples.

Special Populations:

  • Line 134: Please insert "of" immediately following 11%
  • This section changes to using "hormonal contraceptive" whereas the prior sections stated "hormone contraceptive". Either is fine, but please choose one for consistency throughout manuscript
  • Line 155-156: Please add a citation for the statement "Women of reproductive age are at the highest risk for developing a major depressive disorder."
  • Line 163: Change the A of Acid to lower case.
  • Line 201: VTE was previously defined so no need to redefine the abbreviation here
  • Line 206: Consider adding an example of a first or second generation progestin since you provide the drospirenone example for 4th generation.

Counseling on Contraception:

  • Line 243: Missing a period after the reference [5]

Reviewer 2 Report

Thank you for your time and consideration of suggestions.

10. Consider 'pharmacists can provide education on appropriate alternatives'.
